# Physiological Mechanism of Abscisic Acid-Induced Heat-Tolerance Responses to Cultivation Techniques in Wheat and Maize—Review

Zhiqiang Tao [1,†], Peng Yan [1,†], Xuepeng Zhang [2,†], Demei Wang [1], Yanjie Wang [1], Xinglin Ma [1], Yushuang Yang [1], Xiwei Liu [1], Xuhong Chang [1,*], Peng Sui [3,*] and Yuanquan Chen [4,*]

1   Key Laboratory of Crop Physiology and Ecology, Institute of Crop Sciences, Chinese Academy of Agricultural Sciences, Ministry of Agriculture and Rural Affairs, Beijing 100081, China; taozhiqiang@caas.cn (Z.T.); yanpeng01@caas.cn (P.Y.); wangdemei@caas.cn (D.W.); wangyanjie@caas.cn (Y.W.); maxinglin@caas.cn (X.M.); yang-ysh@163.com (Y.Y.); liuxiwei@caas.cn (X.L.)
2   Crop Research Institute, Shandong Academy of Agricultural Sciences, Jinan 250100, China; 13001297977@163.com
3   College of Agronomy and Biotechnology, China Agricultural University, Beijing 100193, China
4   National Academy of Agricultural Science and Technology Strategy, China Agricultural University, Beijing 100193, China
*   Correspondence: changxuhong@caas.cn (X.C.); suipeng@cau.edu.cn (P.S.); chenyq@cau.edu.cn (Y.C.); Tel./Fax: +86-10-8210-8576 (X.C.); +86-10-6273-1163 (P.S.); +86-10-6273-1163 (Y.C.)
†   These authors contributed equally to this work.

**Abstract:** Abscisic acid (ABA) plays a physiological role in regulating the heat tolerance of plants and maintaining crop productivity under high-temperature stress. Appropriate cultivation techniques can regulate endogenous ABA and help farmers improve food production under high-temperature stress. Here, the physiological basis for ABA-induced heat tolerance in crops is reviewed. High-temperature stress stimulates ABA, which reduces stomatal opening and promotes root growth. The root system absorbs water to maintain the water status, thus allowing the plant to maintain physiological activities under high-temperature stress. ABA plays a synergistic role with nicotinamide adenine dinucleotide biosynthesis to improve the thermal stability of the cell membrane, maintain a dynamic balance between material and energy, and reduce the negative effects of high-temperature stress on kernel number and kernel weight. Cultivation and tillage techniques adapted to high-temperature stress, such as adjustment of sowing time, application of plant growth regulators and fertilizers, and the use of irrigation, subsoiling and heat acclimation, and the mechanisms by which they improve crop heat tolerance, are also reviewed. The results of the studies reviewed here will help researchers develop techniques for cultivating food crops under heat stress and apply them to food-production fields to improve crop productivity.

**Keywords:** Abscisic acid; heat stress; crop management; wheat

## 1. Introduction

Abscisic acid (ABA) is an important hormone in crops and plays a role as a signaling molecule in biosynthesis, catabolism, transport and signal transduction pathways. ABA synthesis in the root plays a major role in regulating root growth and the absorption of soil water [1]. ABA also plays a key role in establishing a hydrophobic barrier to prevent water loss [2]. Under high-temperature stress and low-water status, ABA synthesized in leaf vascular cells can be transferred to guard cells to induce stomatal closure, reducing water loss through transpiration [3]. This reduces the amount of $CO_2$ that enters the blade, limiting photosynthesis but reducing water loss; this allows the plant to maintain physiological functions, which improves heat resistance and reduces production losses [4–6]. In addition, in response to high-temperature stress, ABA can enhance glucose metabolism and provide

energy for heat resistance [7]. ABA can also increase the supply of assimilates to ears, enhancing the fertility of pollen, increasing kernel number, and improving grain filling [8–10]. ABA induces the production of heat-shock proteins (*HSPs*), which accumulate in the cell membrane and protect the membrane against thermal denaturation and damage from reactive oxygen species (ROS) [11,12]. ABA also induces the production of energy in the form of adenosine triphosphate (ATP), which is needed for *HSP* accumulation [13]. Several approaches have been taken to improve crop production under heat stress, such as selecting varieties with improved heat tolerance [14], adjusting the sowing time [15], applying plant growth regulators [16] and different fertilizers [17–21], optimizing irrigation systems [22], subsoiling soil [23] and employing heat acclimation. However, although cultivation and tillage techniques have been used by farmers to improve the heat tolerance of crops and scientific researchers have integrated multiple techniques to develop a system that can allow crops to adapt to high-temperature stress, the mechanisms by which cultivation techniques improve heat tolerance by regulating ABA levels have not been summarized. This review aims to summarize the mechanisms of stomatal closure, root growth promotion, plant water maintenance, glucose metabolism regulation and the supply of photosynthate and energy to ear organs induced by ABA under high-temperature stress, and discusses how cultivation and tillage techniques can improve crop heat tolerance by regulating ABA. The purpose is to provide a theoretical basis for the development of coping strategies for wheat and maize production under high temperature from the perspective of crop cultivation and tillage techniques.

## 2. Improved Heat Resistance through ABA-Induced Stomatal Closure, Promotion of Root Growth, and Maintenance of Plant Water Status

### 2.1. Hydrogen Sulfide Promotes ABA-Induced Stomatal Closure

Hydrogen sulfide ($H_2S$) is a gaseous chemical messenger in plants that activates or inhibits ABA-induced stomatal closure and plays an important physiological role in the adaptation of crops to abiotic stress [24] (Figure 1). Upon ABA-induced desulfurization, cysteine desulfurase degrades cysteine and produces $H_2S$ [25–28]. $H_2S$ enhances the persulfidation of SNF1-RELATED PROTEIN KINASE2.6 (SnRK2.6) to promote ABA-induced stomatal closure [29]. This process occurs in the guard cell, and at the same time ABA induces the production of ROS, which destroys the guard-cell membrane, thus affecting the normal opening and closing of stomata. This damage is mitigated by the biosynthesis of nicotinamide adenine dinucleotide, which regulates the activity of nicotinate/nicotinamide mononucleotide adenyltransferase to prevent ABA-induced ROS from harming guard cells [30]. ABA-induced persulfidation of SnRK2.6 also promotes cytosolic calcium ion ($Ca^{2+}$) signaling [29]. $Ca^{2+}$ flows and accumulates in guard cells [31,32]. $Ca^{2+}$ acts as a second messenger in the ABA signaling pathway in guard cells, inhibiting $K^+$ channels while activating anion channels; this causes the osmotic pressure of the guard cells to drop rapidly and water to flow out, leading to stomatal closure [33]. $H_2S$ also promotes the accumulation of nitric oxide (NO) [24], which inhibits ABA signaling in guard cells by inhibiting SnRK2.6 through S-nitrosylation [34,35].

### 2.2. ABA Promotes Root Growth and Soil Water Absorption, Which in Turn Maintains the Plant Water Status

In root meristems, ABA inhibits cell division in the quiescent center, which is the source of stem cells in the roots, and inhibits the differentiation of stem cells and their daughter cells around the quiescent center, thus promoting root growth. In contrast, ethylene promotes cell division in the quiescent center [36]. ABA has been shown to inhibit the expression of ethylene biosynthesis genes in roots, thus inhibiting cell division [36–38]. This promotes the growth and elongation of primary and lateral roots [38,39] and increases the absorption of soil water [1,40], ensuring that plants meet their water requirements (Figure 1).

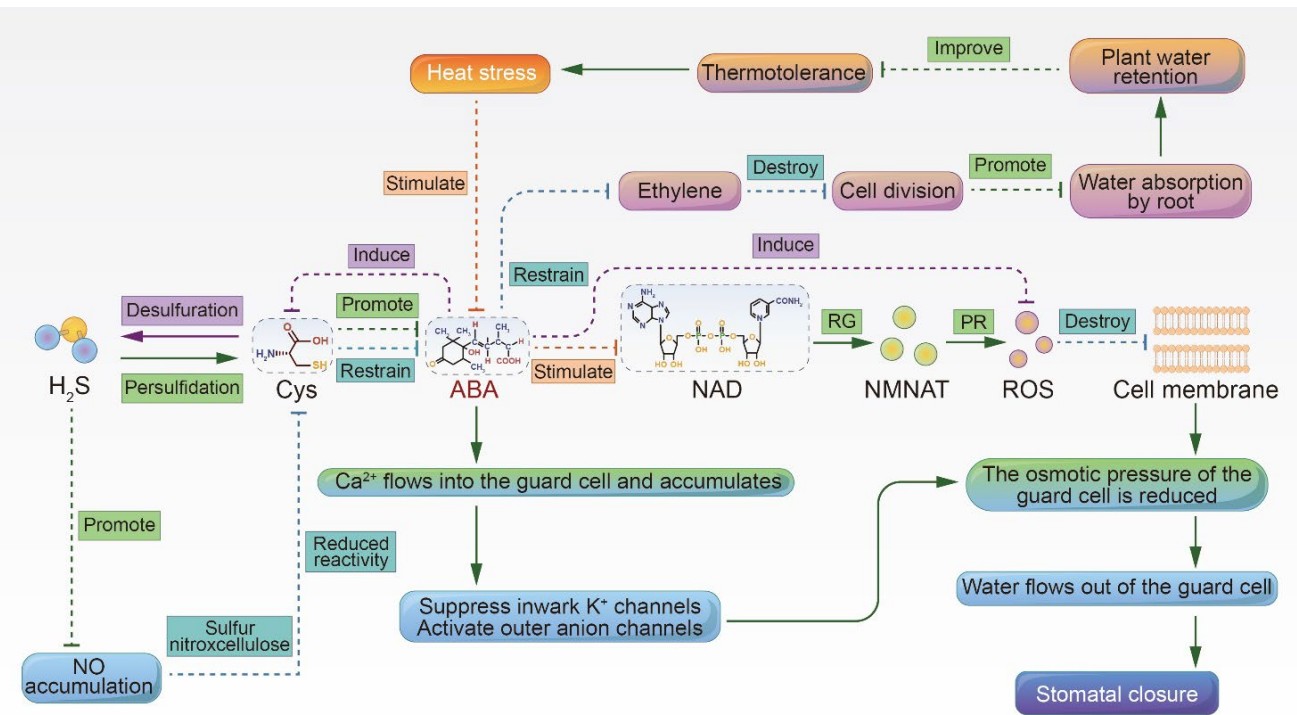

**Figure 1.** Diagram showing the mechanisms by which ABA improves heat resistance by inducing stomatal closure, promoting root growth and maintaining plant water status. ABA, abscisic acid; Cys, cysteine; $H_2S$, hydrogen sulfide; NAD, nicotinamide adenine dinucleotide biosynthesis; NMNAT, nicotinate/nicotinamide mononucleotide adenyltransferase; NO, nitric oxide; PR, prevent; RG, regulate; ROS, reactive oxygen species.

## 3. ABA Regulates Sugar Metabolism in the Spike and Improves Heat Tolerance

### 3.1. ABA Enhances Sucrose Transport and Metabolism

After carbon dioxide ($CO_2$) passes through stomata and enters chloroplasts, it is assimilated into sugar, which is the plant's main source of energy (Figure 2) [41,42]. High-temperature stress reduces carbon metabolism and the amount of carbon assimilated into sugar, resulting in insufficient energy to maintain pollen to resist high temperatures, which in turn results in abnormal pollen development and spikelet abortion [7]. In rice panicles, there is a decrease in the contents of starch and hexose, the activities of sucrose phosphate synthase (SPS) and sucrose synthase, and the content of sucrose, leading to a decline in pollen function, decreased pollination and setting capacity, and an increased abortion rate [43,44]. In rice, ABA enhances sucrose transport and metabolism, which in turn provides the energy pollen needs to cope with high temperatures [7,45]. Under high-temperature stress, increasing ABA content in panicles and ears increases pollen fertility [9]. In one study, the negative effects of high-temperature stress on the contents of nonstructural carbohydrates, soluble sugars (sucrose, glucose and fructose), and starch and dry-matter accumulation in grains were alleviated after treating rice panicles with 100 μmol L$^{-1}$ ABA; pollen viability under high-temperature stress was also improved, and the number of sterile spikelets was reduced. Under high-temperature stress, ABA enhances sucrose unloading and metabolism in the panicle and promotes the redistribution of dry matter from the stem, sheath and leaf to the panicle, ensuring that the photosynthate demand required for grain filling is met [8].

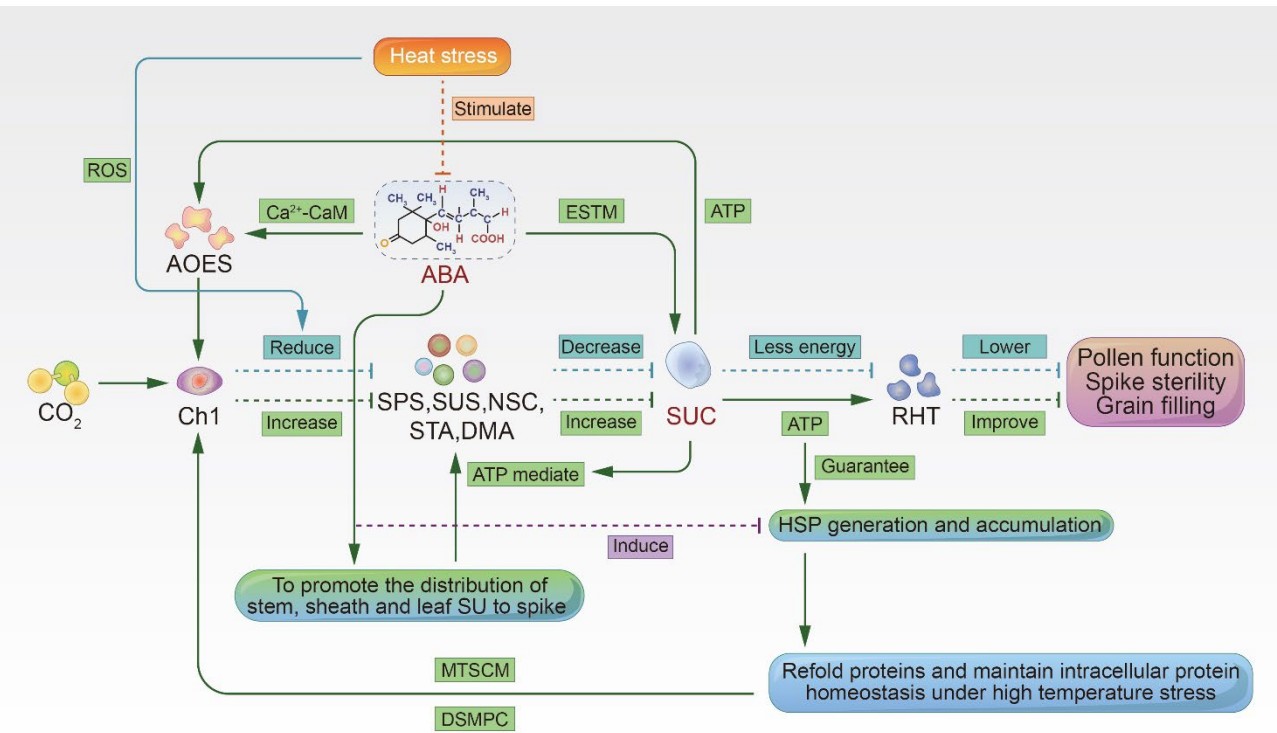

**Figure 2.** Mind map showing how ABA regulates sugar metabolism and improves heat tolerance in plants. AOES, antioxidant defense; ATP, adenosine triphosphate; Chl, chloroplast; DM, dry matter; DMA, dry-matter accumulation; DSMPC, delayed senescence and maintenance of photosynthetic capacity; ESTM, enhanced sucrose transport and metabolism; *HSPs*, heat-shock proteins; MTSCM, maintaining the thermal stability of the cell membrane; NSC, nonstructural carbohydrate; RHT, resist high temperature; ROS, reactive oxygen species; SPS, sucrose phosphate synthase; STA, starch content; SU, sucrose; SUC, sucrose content; SUS, sucrose synthase.

### 3.2. ABA Induces Antioxidant Defenses in Cells

High-temperature stress results in the production of many reactive oxygen species (ROS) (e.g., HO, RO, $O_2$, $HO_2$, ROO, and $^1O_2$) and causes membrane lipid peroxidation, increased malondialdehyde content and rubisco degradation (Figure 2) [46,47]. The activities of antioxidant enzymes (superoxide dismutase, peroxidase, catalase, and ascorbic acid reductase) under ABA treatment are increased, the production of malondialdehyde is decreased, and high-temperature damage is reduced or prevented [48]. This ABA-induced antioxidant defense process, which involves $H_2O_2$ acting as a signal and $Ca^{2+}$-calmodulin-dependent protein (CaM), reduces or prevents ROS oxidation, which destroys biomolecules [11,49]. Glucose is also involved in antioxidant defenses because it provides a source of energy for the biosynthesis of antioxidants through the oxidative pentose phosphate pathway [50].

### 3.3. ABA Induces the Accumulation of Heat Shock Proteins

ABA induces the production and accumulation of *HSPs*, and both ABA-dependent and -independent pathways regulate *HSPs* expression (Figure 2), which plays an important role in improving the heat tolerance of crops [12,51,52]. Under high-temperature stress, cell proteins become unfolded or misfolded and then lose their catalytic structures and activities. As molecular chaperones, *HSPs* participate in protein metabolism, folding and translocation; they improve heat resistance by restoring protein conformation, maintaining protein homeostasis in cells, preventing or reducing high temperature stress-induced damage to the thylakoid membrane and other cell membranes, and maintaining the normal functions of chloroplasts and other organs [13,53].

### 3.4. Adenosine Triphosphate Provides Energy to Support the Accumulation of HSPs

Respiration and metabolic intensity increase under high-temperature stress, and large amounts of carbohydrates and ATP are consumed [54]. ABA can inhibit respiration and increase metabolic capacity [55–57], reducing carbohydrate and ATP consumption to supply energy for the accumulation of *HSPs*, which play crucial roles as molecular chaperones under high-temperature stress [58–60]. High-temperature stress restricts the supply of ATP and inhibits sugar transport from leaves, stems, and sheaths to spikelets via an apoplastic pathway, but ABA promotes the production of ATP in the spike by activating sucrose, which breaks down sucrose into glucose to produce ATP (Figure 2).

## 4. The Mechanism of How Cultivation Techniques Respond to High Temperature in Wheat and Maize

### 4.1. Cultivars

One approach for improving production under heat stress is to select cultivars with specific characteristics. For example, heat cultivars adapted to high temperatures have the following characteristics: (i) The heading time is advanced. Wheat varieties with early heading were selected in North and South China, France and Australia to facilitate adaptation to high-temperature stress during the grain-filling period caused by climate warming [61,62]. (ii) Organs have a strong antioxidant capacity. When wheat is subjected to high-temperature stress, leaves and non-leaf organs (flag leaf sheath, internodes under the ear, glumes and kernels) have strong and sustained antioxidant activity, and the antioxidant capacity of non-leaf organs is higher than that of leaves [63]. (iii) Starch-synthesis-related enzyme activity is high. The activities of ADP-glucose-pyrophosphorylase, a starch branching enzyme and sucrose synthase in wheat seeds, were significantly positively correlated with the contents of amylopectin and total starch in grains at the late stage of grain filling (22–26 d after flowering) [64]. Increasing the activity of enzymes involved in the synthesis of these sugars or starches can improve heat resistance, increase the starch content in the grains, and reduce the effect of high temperature on yield [65]. (iv) The capacity for storage protein accumulation is high, which can compensate for the loss of yield caused by the decrease in starch accumulation [66–68]. One study found that the content of protein and its components was increased by high-temperature stress at the grain-filling stage [69] and that the activity of enzymes related to starch synthesis decreased, but the levels of starch precursors remained relatively stable.

The characteristics of maize varieties adapted to high temperatures are as follows: (i) The heading time of maize is advanced. When maize was subjected to high-temperature stress (40 °C day/30 °C night) before flowering, heading occurred earlier than when maize was grown under normal temperatures (32 °C day/22 °C night) [70]. (ii) Pollen function is not affected or is only slightly affected. One study found that under high-temperature stress, the morphology of pollen in the male tassels of a heat-tolerant cultivar was not deformed as it was in a heat-sensitive cultivar, and the conversion of sugar into starch in the pollen was not disrupted. In addition, the number and size of starch particles in the pollen of the heat-tolerant cultivar did not decrease with increasing temperature, resulting in normal or slightly lower pollen viability and lower pollen shedding compared with the heat-sensitive cultivar. High temperature had no effect on silking in the heat-tolerant cultivar and did not prolong the interval between flowering and silking as it did in the heat-sensitive cultivar; therefore, no decrease in pollination was observed [70]. (iii) Starch-synthesis-related enzyme activity is high. Compared with heat-sensitive cultivars, the activity of starch-synthesis enzymes and starch content were higher in heat-tolerant maize cultivars during grain filling under heat stress, and the decrease in cultivars producing inferior grain was lower than that in those producing superior grain [71]. Compared with a heat-sensitive cultivar, under high-temperature stress, the activities of the synthesis of sucrose phosphate synthase and sucrose synthase were higher, and there was lower sucrose decomposition and higher sugar content in the heat-tolerant cultivar, resulting in more photosynthate supply and a longer grain-filling process [72]. (iv) Under high-

temperature stress, heat-tolerant maize cultivars can maintain higher root activity, a higher leaf photosynthetic rate, higher antioxidant capacity, and higher activity of ATP-producing enzymes in grains than heat-sensitive cultivars [73].

Figure 3 illustrates how wheat and maize cultivars adapt to high temperature; this figure combines the information in Figures 1 and 2 with the characteristics of heat-tolerant cultivars. Under high-temperature stress, ABA promotes root growth, and roots absorb water to maintain the water status of the plant. Roots also absorb nitrogen to supply the requirement to produce storage proteins in the grains. ABA also regulates stomatal opening and closing, maintains photosynthesis, and promotes starch and sugar synthesis to meet the demand of the panicle for photosynthates and energy. ABA induces the accumulation of the molecular chaperones *HSPs*, which improve the antioxidant capacity of chloroplasts and maintain photosynthesis. Storage proteins also compensate for the loss of starch in the grains. A sufficient amount of photosynthates and energy guarantees pollination and a high kernel number per ear, increases grain filling and kernel weight, and weakens the inhibitory effect of high temperature on yield.

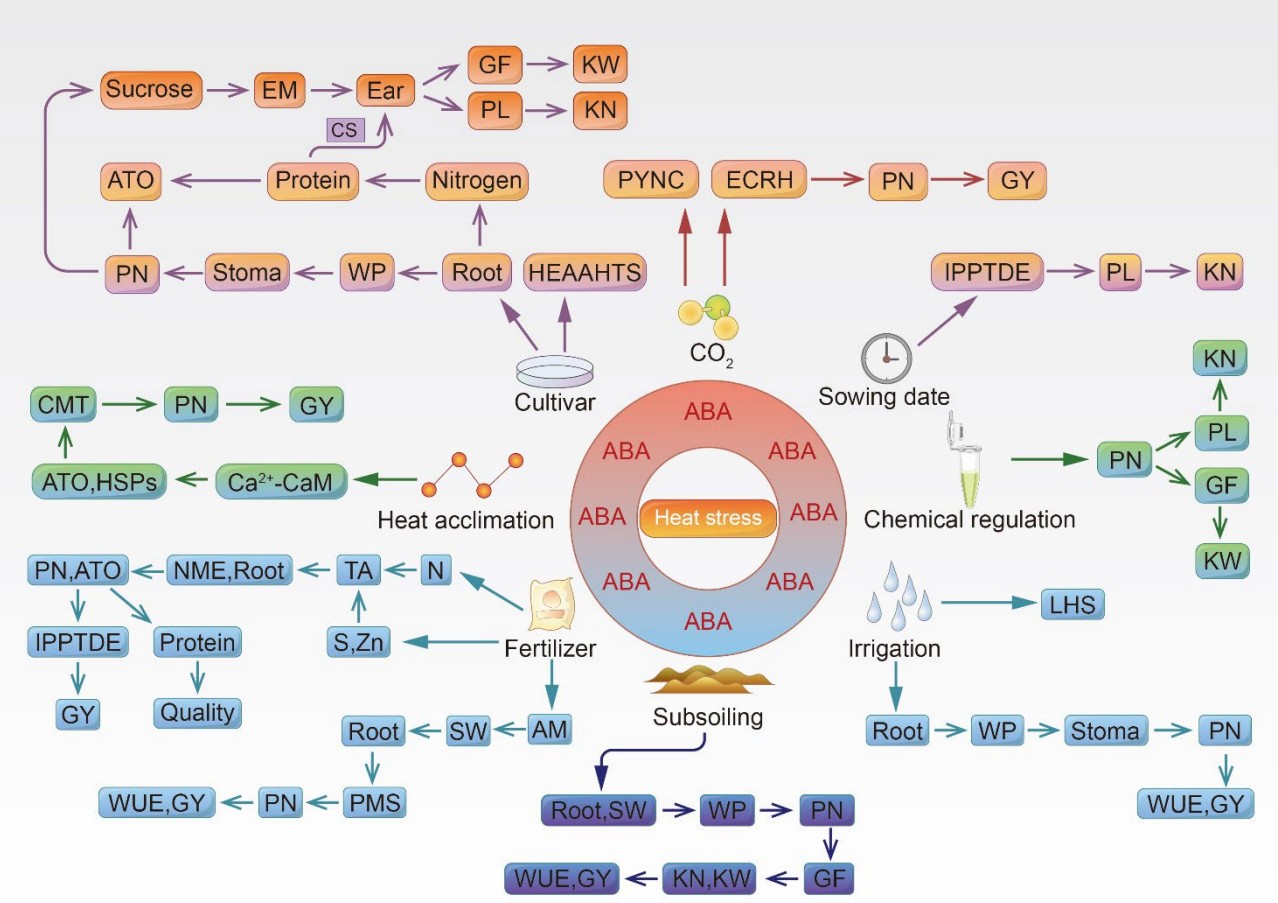

**Figure 3.** Diagram of how cultivation and farming techniques facilitate the adaptation of wheat and maize to high-temperature stress. AM, arbuscular mycorrhizal fungi; ATO, antioxidant ability; CMT, cell-membrane thermostability; CS, compensation for the loss of starch; ECRH, equilibrium of cell redox homeostasis; EM, energy and matter; GF, grain filling; GS, glutamine synthetase; GY, grain yield; HEAAHTS, heading early to avoid or adapt to high-temperature stress; IPPTDE, improve the production of photosynthates and promote transfer and distribution to ears; KN, kernel number per ear; KW, kernel weight; LHS, lower heat sensitivity; NME, nitrogen metabolism enzyme; PL, pollination; PMS, plant moisture status; PN, photosynthesis; PYNC, protein yield negatively correlated with $CO_2$ concentration; SW, soil water-holding capacity; TA, topdressing before and after anthesis; WP, water status of the plant; WUE, water use efficiency.

### 4.2. Sowing Date

The sowing date of wheat or maize can be adjusted to affect the amount of light and heat the crop is exposed to, which affects the ABA contents in the stem, leaves, and ears of the plant. Optimizing ABA content can improve the production of photosynthates and promote the transport and distribution of photosynthates to ears, which ensures pollination and seed setting and increases kernel number per ear (Figure 3). There is a linear relationship between wheat yield and grain number per unit area, and floret fertility is greatly affected under high-temperature stress. In one study conducted in Taian City, Shandong Province, China, wheat cultivars sown on 15 October or 22 October were compared with those sown on 1 October or 8 October. Cultivars sown at the later date had higher ABA contents in the stem and lower ABA contents in the panicle, which promoted the transport of carbon compounds and water-soluble carbohydrates to the panicle and led to an increased number of florets and grains per panicle [15]. In Cangzhou city, Hebei Province, China, spring maize sown in late May was compared with that sown from early April to mid-May. During the 15 days before and after silking for maize, there were fewer days of high-temperature stress, which was correlated with a relatively low ABA content, higher amount of dry-matter accumulation and an increased number of grains [74].

### 4.3. Chemical Regulation

Plant growth regulators that are commonly applied to wheat and maize to promote resistance to high-temperature stress are 2,4-dichloroformamide cyclopropane acid (2,4-D) and monopotassium phosphate ($KH_2PO_4$). These chemicals improve photosynthesis, reduce the length of the tip of a maize cob, increase kernel number per ear, and increase the grain-filling rate and kernel weight under high-temperature stress (Figure 3). When 2,4-D was sprayed on the leaves at the ninth-expanded-leaf stage of spring maize grown under heat stress, the ear leaf had a significantly increased chlorophyll content and ABA content and an enhanced net photosynthesis rate at the grain-filling stage compared with plants that were not sprayed. Spraying also improved the grain-filling rate, grain volume, and grain dry weight and decreased the bare tip length, which increased the grain yield by 8.5% [16].

In wheat plants sprayed with $KH_2PO_4$, compared with those not sprayed with $KH_2PO_4$, the ABA content 4–20 d after flowering and the ethylene evolution rate 4–16 d after flowering were significantly higher in inferior grains (the grains in the middle spikelets (spikelets 4–12) most distal from the bottom of the spike), and the ratio of ABA/ethylene was also higher. Moreover, the maximum and mean grain-filling rates and grain weight were significantly and positively correlated with the ratio of ABA/ethylene in the inferior grains. As the inferior grain weight was significantly and positively correlated with the ear kernel weight, this is the main reason why there was an increase in kernel weight in wheat sprayed with $KH_2PO_4$ [75].

### 4.4. Fertilization

#### 4.4.1. Nitrogen Fertilization

The rate of nitrogen application and whether soil water is sufficient can affect the degree of heat tolerance [76,77]. During the production of wheat in Shijiazhuang city in the North China Plain, infrared heaters were used to increase the ambient temperature. Compared with fields without nitrogen fertilizer, in fields receiving an application of nitrogen fertilizer (240 kg ha$^{-1}$), the soil temperature in the 5–40 cm soil layer increased by 0.2–0.3 °C, the soil volumetric water content in the 0–60 cm soil layer decreased by 0–3.4%, and the yield increased significantly ($p < 0.05$), but the reduction in yield under heat stress was larger for crops receiving nitrogen fertilizer than for those that did not. This is mainly because, in the nitrogen fertilizer treatment, evapotranspiration increased with increasing air temperature and soil temperature, which caused a decrease in the soil water content and a significant decrease in the number of panicles [77]. This shows that nitrogen fertilizer helps improve crop productivity under heat stress when there is adequate soil

moisture. In addition, the results of several studies suggest that farmers should control the total amount of nitrogen fertilization. In Catalonia, in northeastern Spain, wheat was subjected to high-temperature stress during the grain-filling period, and the amount of nitrogen applied to the field increased from 68 kg ha$^{-1}$ to 376 kg ha$^{-1}$; yield loss, which ranged from 10–25%, increased with increasing nitrogen applied [78]. In a study of fresh waxy maize, compared with nitrogen application rates of 1.5 and 7.5 g plant$^{-1}$, nitrogen application rates of 4.5 g plant$^{-1}$ alleviated the negative effects of high-temperature stress (35 °C day/21 °C night) on grain yield and quality [79]. High-temperature stress accelerates the leaf senescence process [80]; a study of maize found that the application of nitrogen at a level of 240 kg ha$^{-1}$ may be a beneficial strategy to improve grain yield by regulating ABA levels and flag leaf senescence [81]. The regulation of ABA levels may also improve starch assimilation, increase the filling rate and prolong the duration of grain filling [82].

The timing of fertilizer application can affect the degree of heat tolerance. Under high-temperature stress, compared with plants treated with all nitrogen fertilizer applied at once as a basal fertilizer, plants treated with half of the nitrogen fertilizer applied as a basal fertilizer and the other half applied as a topdressing at the jointing stage or booting stage had increased glutamine synthetase, catalase and peroxidase activity in the flag leaf, increased photosynthetic rate and stomatal conductance, higher translocation of dry matter from the vegetative organs before flowering to grain, improved heat tolerance, and significantly increased kernel weight and grain yield [83]. In another study of spring maize in Cangzhou City of the North China Plain, before being subjected to heat stress during the grain-filling stage, plants were treated with a single application of nitrogen as a base fertilizer or four applications of nitrogen fertilizer (total of 300 kg ha$^{-1}$) at the following ratio: planting:V7 (seven leaves with collar visible):V15 (15 leaves with collar visible):R3 (silks dried out, approximately 20 days after silking) = 60:90:60:90 kg ha$^{-1}$. The 60:90:60:90 kg ha$^{-1}$ treatment reduced the electrical conductivity (the heat tolerance of plants is related to cell-membrane thermostability, which is commonly estimated by measuring the electrical conductivity of plant segments after a defined heat treatment) and the malondialdehyde content of the ear leaves and increased the chlorophyll content, net photosynthetic rate of the ear leaves (by 9.95%), kernel number per ear, and yield (by 13.47%); ABA was found to be involved in regulating antioxidant processes in flag leaves [84]. Another study found that, compared with applying a nitrogen topdressing to Spanish maize before and after flowering, broadcasting N urea at V6 (six leaves with collar visible) at a rate of 200 kg ha$^{-1}$ increased the ear temperature by 1 °C from the silking stage to the grain-filling stage, reduced the kernel number per ear, and increased the extent of the yield reduction [85].

### 4.4.2. Sulfur Fertilization

Sulfur (S) is a mineral nutrient essential for plant growth. It is the fourth major nutrient after nitrogen, phosphorus and potassium. Ninety percent of the S in plants is used to synthesize S-containing amino acids. In recent years, the use of S in industrial production processes has decreased, resulting in reduced atmospheric acid deposition and the use of high-concentration phosphate fertilizers with less S; therefore, S has become a major constraint on crop production [86,87]. When oil-seed rape plants treated with low S (8.7 μM SO$_4{}^{2-}$) or high S (500 μM SO$_4{}^{2-}$) during the seed-filling stage were subjected to high-temperature stress (33 °C/day, 19 °C/night vs. control temperature 20 °C/day, 15 °C/night), high-temperature stress and low S decreased the number of seeds per plant; decreased the ABA content in grains; decreased the (raffinose + stachyose)/sucrose ratio, linoleic acid/linolenic acid ratio, and S-poor seed-storage-protein (12S albumins)/S-rich seed storage protein (2S albumins) ratio. This led to impaired synthesis of methionine and cysteine. The balance of protein production was maintained, but the grain yield and quality were reduced [88].

Supplementary application of S fertilizer can improve the heat resistance of crops and improve grain yield and quality. For example, it was found that high-temperature stress

significantly increased the contents of total protein, albumin, alcohol soluble protein, gluten, cysteine and methionine in wheat grains ($p < 0.05$) but decreased grain yield, grain weight, globulin content, total starch accumulation and overall quality. However, supplementary application of S fertilizer at the heading stage had a significant positive effect on the activities of nitrate reductase and other nitrogen-metabolism enzymes, glucose metabolism and photosynthesis [87]. S application improved nitrate reductase and glutamine synthetase activities in the flag leaf, grain yield, grain weight, and the contents of cysteine, methionine, total protein, albumin, alcohol-soluble protein, gluten and globulin and improved the overall quality [89]. These results show that the application of S fertilizer alleviates the negative effects of high-temperature stress on grain yield, starch content and grain quality.

### 4.4.3. Zinc Fertilization

Foliar application of zinc (Zn) fertilizer or related nutrient compounds has an obvious effect on the resistance of wheat and corn to abiotic stress. Treating soil with 15 mg Zn kg$^{-1}$ was found to increase the grain yield, protein yield, grain weight, and total protein, albumin, gliadin and gluten contents, as well as the activities of nitrate reductase and glutamine synthetase in flag leaves after 20 days of high-temperature stress at the filling stage of wheat, but decreased the globulin content; the negative effects of high-temperature stress on grain yield, protein content, protein component content and comprehensive quality were also reduced [20]. The addition of chitosan oligosaccharides and marine polysaccharides to nutritional compound preparations composed of Zn sulfate, potassium dihydrogen phosphate and urea was found to effectively enhance root activity, increase the chlorophyll content and betaine content of the flag leaf, delay leaf senescence, improve grain filling, and increase the 1000-kernel weight and the harvest index in wheat; in general, the harm caused by heat stress during grain filling in wheat was reduced, and the yield was stabilized [90]. In the dry land of western Henan, China, increased Zn fertilizer application was found to enhance root stability under drought stress, improve the development of wheat roots and alleviate the effect of drought stress on yield reduction [91]. Applying Zn fertilizer to maize in Henan Province increased the leaf area, chlorophyll content, net photosynthetic rate, total amount of dry matter per plant, accumulation of dry matter in each organ, and transport of dry matter to seeds, and enhanced drought resistance and yield [92].

### 4.4.4. Arbuscular Mycorrhizal Fungi Fertilization

Arbuscular mycorrhizal fungi application can increase heat tolerance. Maize roots inoculated with arbuscular mycorrhizal fungi under heat stress can improve heat resistance by improving photosynthesis and water status. Arbuscular mycorrhizal fungi help optimize the soil water-holding capacity, which improves the plant moisture status (relative water content in maize leaves) and then indirectly increases stomatal conductance and increases photosynthesis parameters, including the net photosynthetic rate, maximal fluorescence, maximum quantum efficiency of PSII photochemistry and potential photochemical efficiency, and concentrations of chlorophyll *a* and chlorophyll *b* [21,93].

### *4.5. Carbon Dioxide*

The stimulation of grain yield by elevated $CO_2$ in wheat depends on high temperatures. It also relies on the availability of nitrogen nutrient resources. A study of 11 wheat varieties in four continents and eight countries found that, for nitrogen application rates up to 200 kg ha$^{-1}$, increasing the $CO_2$ concentration can increase grain yield. However, for nitrogen application rates >200 kg ha$^{-1}$, when the $CO_2$ concentration was increased, grain yield stagnated or even decreased. For all nitrogen application rates, increasing $CO_2$ significantly reduced grain protein yield by 7% on average. This suggests that, in an environment where extreme temperatures are common because of global warming, studying the interaction between nitrogen fertilizer and $CO_2$ is an effective way to simultaneously improve wheat yield and quality. Plant responses to increased $CO_2$ and higher temperatures are regulated by ABA and redox homeostasis networks [94]. Increased $CO_2$ concentrations

balance cellular redox homeostasis and increase malondialdehyde content and electrolyte leakage but also significantly increase the antioxidant capacity of plants, improving the Fv/Fm (the ratio of variable to maximum fluorescence after dark adaptation, representing the maximum quantum yield of photosystem II) of plants subjected to high-temperature stress [95]. ABA is indirectly involved in the mitigation of high-temperature stress induced by increased $CO_2$ concentrations by increasing the antioxidant capacity of the plants.

*4.6. Irrigation*

Irrigation has become an important method for adapting global crop production to climate change. In India, the heat sensitivity of irrigated wheat was found to be only one-quarter of the heat sensitivity of wheat grown under rain-fed conditions. However, as the negative effects of climate change continue and additional constraints on expanding irrigation are imposed, increasing production through irrigation in a warming climate will be a serious challenge [96].

In China, which has limited water resources, scientists and agricultural production operators are improving irrigation technologies to cope with the negative impacts of climate change on crop production. In North China, the use of micro-sprinkling hoses (5–10 mm) during the grain-filling stage of wheat at 10:00 on days with forecasted high temperatures can significantly reduce the canopy temperature and increase the relative humidity of the canopy, the water potential of the flag leaf and the photosynthetic rate of the plant population. Furthermore, the earlier the time of micro-sprinkling, the higher the increase in kernel weight and grain yield [97]. In semiarid regions of China, it was found that supplementary irrigation of 30 mm during the flowering stage of wheat could prevent future yield losses due to the increase in $CO_2$ concentrations and temperatures caused by climate change. Higher supplemental irrigation levels of 60 and 90 mm were found to increase wheat yields by 3.8 and 10.1%, respectively. Thus, supplementary irrigation (30–90 mm) may play a key role in maintaining rainfed spring wheat yields in regions affected by global climate change [98].

A study of supplemental irrigation under high-temperature stress during the rice heading and filling period in South China found that, compared with water layer irrigation, light dry and wet alternate irrigation (rehydration when the soil dries to a soil water potential of $-15$ kPa) reduced the relative humidity of the canopy, increased the endogenous ABA content in leaves, reduced the rate of ROS generation in leaves, increased the levels of the antioxidants ascorbic acid and reduced glutathione, increased the concentrations of endogenous cytokinins and spermine, delayed leaf senescence, and significantly increased the seed setting rate, 1000-kernel weight and grain yield [22]. An alternative irrigation technique, controlled root-divided alternative irrigation, has been widely studied and applied in wheat and maize production to maintain food production; the use of this method, which promotes the transport of ABA from the roots to the leaves where it regulates stomatal opening, can greatly reduce luxury transpiration without sacrificing the accumulation of photosynthetic products [99,100]. Therefore, we speculate that the use of alternative irrigation technologies may be an important adaptive measure for maintaining food-crop production under high-temperature stress.

*4.7. Subsoiling*

To cope with high-temperature stress during the spring maize grain-filling period in the North China Plain, subsoiling is performed before planting. Subsoiling can directly increase the root length density and soil moisture in the 0–80 cm soil profile, indirectly alleviate the inhibitory effects of high temperature on leaf photosynthetic rate and plant water status, increase the ABA-induced activity of superoxide dismutase, decrease the content of malondialdehyde, increase heat tolerance, improve the filling rate, prolong the grain linear-filling stage, and increase grain number per spike and yield [23,101].

*4.8. Heat Acclimation*

Heat acclimation is also an effective way to increase the heat tolerance of crops. Crops that have been acclimated through high temperatures can maintain low respiratory costs and exhibit no or slight reductions in photosynthesis under high-temperature stress, which allows the crop to maintain net carbon gain [102]. In one study of heat-tolerant and heat-sensitive varieties of wheat seedlings, the thermal death times at 50 °C ranged from 8–26 min. After acclimation for 3 days at 34 °C, the thermal lethal times of heat-tolerant and heat-sensitive varieties at 50 °C increased to 87–110 min and 35–55 min, respectively. The reason for this increase is that heat acclimation increases the activity and stability of superoxide dismutase and catalase, improves the stability of the leaf cell membrane, and stimulates *HSPs*, which increase protein stability; this enhances the antioxidant capacity of the plants under high-temperature stress, prolonging the thermal death time [103,104]. Of these traits, the thermal stability of the cell membrane is considered an ideal physiological index for evaluating heat resistance [105].

The heat response of maize seedlings is initiated through the intracellular entry of extracellular $Ca^{2+}$ and the regulation of intracellular CaM after heat acclimation. Heat acclimation enhances the activity of antioxidant enzymes such as superoxide dismutase, catalase and ascorbic acid reductase, which are induced by ABA, and lowers heat stress-induced lipid peroxidation [46]. The heat-acclimation process in wheat is also accompanied by the accumulation of CaM. Moreover, the accumulation of CaM is affected by the concentrations of $Ca^{2+}$; $Ca^{2+}$-CaM signaling can regulate the heat-shock response through *HSP70* [106].

## 5. Concluding Remarks and Future Perspectives

The adaptation of wheat and maize to high-temperature stress requires appropriate cultivation and farming techniques. We reviewed the physiological basis of heat tolerance induced by ABA under high-temperature stress and the mechanisms underlying ABA-mediated adaptation to high-temperature stress under different wheat and maize cultivation and farming techniques.

High-temperature stress induces ABA, which reduces stoma opening and increases root water-absorbing capacity, which together maintain the water status and the physiological activities of plants under high-temperature stress. High-temperature stress-induced production of ABA also enhances sucrose transport and metabolism and promotes the distribution of dry matter from the stem, sheath and leaf to the panicle in an ATP-dependent manner. The activity of antioxidant enzymes is increased by ABA, and ABA induces the production and accumulation of *HSPs*. Antioxidant enzymes and *HSPs* work together to remove ROS to prevent or mitigate damage to the thylakoid membrane and other cell membranes and to maintain the normal functions of chloroplasts and other organs under high-temperature stress. At the same time, ABA inhibits respiration and increases metabolic capacity, reduces carbohydrate and ATP consumption, maintains a dynamic balance between carbohydrates and energy, and ensures the availability of carbohydrates and energy required for the antioxidant defense system. This system improves pollen function, panicle fertility and grain filling under high-temperature stress and alleviates the negative effect of heat stress on productivity.

Heat-tolerant cultivation and tillage techniques may be used for improving production in wheat and maize under heat stress. The physiological basis for these methods is the regulation of ABA to induce heat tolerance in crops. For example, heat-tolerant wheat and corn varieties with the following characteristics can be chosen: advanced heading time, large antioxidant capacity, high starch-synthesis-related enzyme activity, large amounts of storage-protein accumulation, and pollen and roots that are resistant to heat stress. Other methods include changing the sowing date to avoid high-temperature stress, spraying plant growth regulators on foliage before and after high-temperature stress to improve heat tolerance, and applying nitrogen, zinc, and sulfur fertilizers as topdressing before and after flowering to reduce the negative effect of high-temperature stress on productivity. In addition, inoculating roots with arbuscular mycorrhizal fungi can reduce the negative effect of

high-temperature stress on yield. Increasing $CO_2$ and higher temperatures are regulated by ABA and redox homeostasis networks to mitigate high-temperature stress. Using a suitable irrigation system can also lower the heat sensitivity of plants. Subsoiling soil prior to sowing can alleviate the negative effects of high-temperature stress on yield. These cultivation and farming systems have been applied in the field to improve crop production. However, there is still a lack of research on the integrated application of multiple technologies. In the future, we should focus on the integration of heat-resistant cultivation techniques and innovate heat-resistant agronomic management. In addition, the relationship between high-temperature stress and ABA needs further study; especially, quantification of the association between ABA, physiological activity and heat tolerance is challenging [107]. We can focus on quantifying ABA levels and the production and transport of photosynthates in plant organs, soil water and roots, ROS, antioxidant enzyme activities, *HSPs*, nitrogen-metabolism enzyme activities, chloroplast structure, photosynthesis and water status in leaf or non-leaf photosynthetic organs, pollination, storage-protein accumulation and grain-filling characteristics in the spike under high-temperature stress and determine how they relate to each other. An understanding of these factors and their interactions will allow the development of a system for cultivating plants under high-temperature stress. Furthermore, this study mainly summarized the mechanism of ABA regulation of heat tolerance from the physiological and biochemical levels, and the mechanism from the level of genomics and proteomics is still to be elaborated in the future.

**Author Contributions:** Conceptualization, Z.T., P.Y., X.Z., X.C., P.S. and Y.C.; formal analysis, Z.T., P.Y. and X.Z.; funding acquisition, X.C.; investigation, Z.T., P.Y. and X.Z.; project administration, D.W., Y.W., X.M., Y.Y. and X.L.; Resources, P.S. and Y.C.; supervision, X.C., P.S. and Y.C.; Validation, Z.T., P.Y. and X.Z.; visualization, Z.T., P.Y. and X.Z.; writing—original draft, Z.T., P.Y. and X.Z.; writing—review & editing, Z.T., P.Y., X.Z., D.W., Y.W., X.M., Y.Y., X.L., X.C., P.S. and Y.C. All authors have read and agreed to the published version of the manuscript.

**Funding:** This research was financially supported by China Agriculture Research System of MOF and MARA (CARS-03-16), the National Natural Science Foundation of China (31701387, 32101858), the Key Technologies Research and Development Program of China (2016YFD0300407, 2017YFD0300601), the Innovation Program of CAAS.

**Institutional Review Board Statement:** Not applicable.

**Informed Consent Statement:** Not applicable.

**Data Availability Statement:** Not applicable.

**Conflicts of Interest:** The authors declare no conflict of interest.

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
