# Peer review of "Physiological Mechanism of Abscisic Acid-Induced Heat-Tolerance Responses to Cultivation Techniques in Wheat and Maize—Review"

_agronomy, doi:10.3390/agronomy12071579_

Round 1

Reviewer 1 Report

General:

This paper is nicely drafted review about Physiological mechanism of abscisic acid-induced heat toler-2 ance responses to cultivation techniques in wheat and maize. It will help in decision making processes of selecting lines for addressing heat tolearnce               

Introduction: Nicely written introduction.

Line 41: Reference style should be numbered

Line 50: Please check recent review paper on heat stress on wheat:

Yadav, M.R., Choudhary, M., Singh, J., Lal, M.K., Jha, P.K., Udawat, P., Gupta, N.K., Rajput, V.D., Garg, N.K., Maheshwari, C. and Hasan, M., 2022. Impacts, Tolerance, Adaptation, and Mitigation of Heat Stress on Wheat under Changing Climates. International Journal of Molecular Sciences, 23(5), p.2838.

Section 4.2: Unbold

Figure fonts should be in Palatino Linotype to match the text

Line 70-72: The hypothesis and objectives should be clearly defined. The future implications of this work can be added

Conclusion

I would suggest the authors to have a more supported discussion with references considering the main point: The limitations of breeding or genetic on this and considerations when to apply the studied methodology and then the potential next steps or further investigation to address these limitations. Heat tolerance mechanism includes all genomics, proteomics, agronomic and other , so author should highlight somewhere in conclusion. It should include limitation and future implications.

References: Please double check the style of references and missing one

Author Response

Dear reviewer,

We appreciate for your comments concerning our manuscript (agronomy-1771772). We have revised the manuscript accordingly and responded point by point to your comments as list below. Hopes this will make it more acceptance for publication.

General: This paper is nicely drafted review about Physiological mechanism of abscisic acid-induced heat toler-2 acne responses to cultivation techniques in wheat and maize. It will help in decision making processes of selecting lines for addressing heat tolerance

Point 1: Introduction: Nicely written introduction.

Response 1: Thanks for referee comments. We double checked the introduction section and revised some gramma mistakes and marked as blue font in the manuscript.

Point 2: Line 41: Reference style should be numbered

Response 2: Thanks for referee comments. We changed the reference style as numerical style instead of name &year style according to the journal requirements. All the changes in the manuscript have been marked as blue font.

Point 3: Line 50: Please check recent review paper on heat stress on wheat:

Yadav, M.R., Choudhary, M., Singh, J., Lal, M.K., Jha, P.K., Udawat, P., Gupta, N.K., Rajput, V.D., Garg, N.K., Maheshwari, C. and Hasan, M., 2022. Impacts, Tolerance, Adaptation, and Mitigation of Heat Stress on Wheat under Changing Climates. International Journal of Molecular Sciences, 23(5), p.2838.

Response 3: Thanks for referee comments. We checked the recent review paper on heat stress on wheat, and revised our manuscript as “ABA can also increase the supply of assimilates to ears, enhancing the fertility of pollen, increasing kernel number, and improving grain filling”, and add the citation in the manuscript.

Point 4: Section 4.2: Unbold

Response 4: Thanks for referee comments. We have revised the font-style of section 4.2 as no bold in the manuscript.

Point 6: Figure fonts should be in Palatino Linotype to match the text

Response 6: Thanks for referee comments. We have revised all figures fonts in the manuscript.

Point 7: Line 70-72: The hypothesis and objectives should be clearly defined. The future implications of this work can be added

Response 7: Thanks for referee comments. We revised this section as “This review aims to summarize the mechanisms of stomatal closure, root growth promotion, plant water maintenance, glucose metabolism regulation and supply of photosynthate and energy to ear organs induced by ABA under high temperature stress, and dis-cussed how cultivation and tillage techniques can improve crop heat tolerance by regulating ABA. The purpose is to provide a theoretical basis for the development of coping strategies for wheat and maize production under high temperature which from the per-spective of crop cultivation and tillage techniques.”

Point 8: Conclusion: I would suggest the authors to have a more supported discussion with references considering the main point: The limitations of breeding or genetic on this and considerations when to apply the studied methodology and then the potential next steps or further investigation to address these limitations. Heat tolerance mechanism includes all genomics, proteomics, agronomic and other , so author should highlight somewhere in conclusion. It should include limitation and future implications.

Response 8: Thanks for referee comments. We have rewritten our conclusion as “These cultivation and farming systems has been applied in the field to improve crop production. However, there is still a lack of research on the integrated application of multiple technologies. In the future, we should focus on the integration of heat-resistant cultivation techniques and innovate heat-resistant agronomic management. In addition, the relationship between high temperature stress and ABA needs further study, especially quantification of the association between ABA, physiological activity and heat tolerance is challenging [111]. We can focus on quantifying ABA levels and the production and transport of photosynthates in plant organs, soil water and roots, ROS, antioxidant enzyme activities, HSPs, nitrogen metabolism enzyme activities, chloroplast structure, photosynthesis and water status in leaf or non-leaf photosynthetic organs, pollination, storage protein accumulation and grain filling characteristics in the spike under high-temperature stress and determine how they relate to each other. An understanding of these factors and their interactions will allow the development of a system for cultivating plants under high-temperature stress. Furthermore, this study mainly summarized the mechanism of ABA regulation of heat tolerance from the physiological and biochemical levels, and the mechanism from the level of genomics and proteomics is still to be elaborated in the future.”

Point 9: References: Please double check the style of references and missing one

Response 9: Thanks for referee comments. We have double checked all the references in the manuscript accordingly to the Journal. All the changes in the manuscript have been marked as blue font.

The authors are grateful to academic editor and reviewers for the advice in the manuscript. Thanks so much for your work. It our great pleasure to get your help, improving the quality of the paper.

Regards,

Zhiqiang

Reviewer 2 Report

The work is an excellent try to review the physiological mechanism of ABA addressing wheat and maize crops.

The construction of the MS shall be focussed with the plant's functioning and cultural operation? which one is the priority and why?

There is more focus on the shoot but not roots whereas the introduction starting statements are that ABA on root, root growth and the whole documents did not address the root mechanism for changes in shoots? I think the instruction needs revision according to the MS or otherwise.

ABA is the major hormones affecting mechanism of the normal plant growth with changes in the quality (e.g. protein denature) which did not explained in the MS?

subheading the sowing date is not properly reflects the ABA production mechanism in plants and its effects on altering yield traits?

I suggest that MS can be focussed with on information of climate and/or cultural operation or this should be unidimensional for attracting more specific readers. 

Author Response

Dear reviewer,

We appreciate for your comments concerning our manuscript (agronomy-1771772). We have revised the manuscript accordingly and responded point by point to your comments as list below. Hopes this will make it more acceptance for publication.

The work is an excellent try to review the physiological mechanism of ABA addressing wheat and maize crops.

Point 1: The construction of the MS shall be focused with the plant's functioning and cultural operation? which one is the priority and why?

Response 1: Thanks for referee comments. The construction of the current manuscript mainly included 4 sections, that are introduction, mechanisms referring to ABA improves heat resistance, cultivation and tillage techniques adapted to high-temperature stress, conclusion and future perspectives. As mentioned in the manuscript, ABA plays a physiological role in regulating plant heat tolerance and maintaining crop productively under heat stress. While the physiological mechanism of abscisic acid-induced heat tolerance is the key to understand why agronomic management practices included cultivation and tillage techniques used so far in major grain producing areas, could mitigate high-temperature stress and yield penalty. Moreover, the studies reviewed in the current manuscript will help researchers develop techniques for cultivating food crops under heat stress and apply them to food production fields to improve crop productivity.

Point 2: There is more focus on the shoot but not roots whereas the introduction starting statements are that ABA on root, root growth and the whole documents did not address the root mechanism for changes in shoots? I think the instruction needs revision according to the MS or otherwise.

Response 2: Thanks for referee comments. We agree with the referee’s advice and revised the introduction section as “ABA synthesis in the root plays a major role in regulating root growth and the absorption of soil water”. In addition, we add detailed mechanism referring to ABA promoting root growth and soil water absorption in section 2.2 as “In root meristems, ABA inhibits cell division in the quiescent center, which is the source of stem cells in the roots, and inhibits the differentiation of stem cells and their daughter cells around the quiescent center, thus promoting root growth. In contrast, ethylene promotes cell division in the quiescent center[1]. ABA has been shown to inhibit the expression of ethylene biosynthesis genes in roots, thus inhibiting cell division[1-3]. This promotes the growth and elongation of primary and lateral roots[4,5] and increases the absorption of soil water[6,7], ensuring that plants meet their water requirements (Fig. 1).” in the manuscript.

Point 3: ABA is the major hormones affecting mechanism of the normal plant growth with changes in the quality (e.g. protein denature) which did not explained in the MS?

Response 3: Thanks for referee comments. We have revised the manuscript and added “ABA induces the production and accumulation of HSPs, and both ABA-dependent and -independent pathways regulate HSPs expression (Fig. 2), which play an important role in improving the heat tolerance of crops[8,9].” in the manuscript.

Point 4: subheading the sowing date is not properly reflects the ABA production mechanism in plants and its effects on altering yield traits?

Response 4: Thanks for referee comments. We agree with the referee’s advice. Basically, sowing date is one of the most important management practices for mitigating or even avoiding high temperature stress for field crops. For example, field crops susceptibility to high temperature stress varies widely among crop growth stages, of which crops showed much more sensitive to heat stress during flowering time than other growth stages. In this case, the sowing date treatments could provide an ideal model for the study of crops response to high temperature stress and the relevant mechanism including ABA production. Therefore, sowing date, for the most part, is an important and effective agronomic method adapt to heat stress, and our advice is kept it here in “The mechanism of how cultivation techniques respond to high temperature in wheat and maize” section in the manuscript.

Point 5: I suggest that MS can be focused with on information of climate and/or cultural operation or this should be unidimensional for attracting more specific readers.

Response 5: Thanks for referee comments. We have revised the cultural operation section in the manuscript to make it more unidimensional for attracting more specific readers.

Point 6: gramma mistake that tagged in the manuscript.

Response 6: Thanks for referee comments. We have revised the gramma mistakes throughout the manuscript.

The authors are grateful to academic editor and reviewers for the advice in the manuscript. Thanks so much for your work. It our great pleasure to get your help, improving the quality of the paper.

Regards,

Zhiqiang

References

  1. Zhang, H.M.; Han, W.; De Smet, I.; Talboys, P.; Loya, R.; Hassan, A.; Rong, H.L.; Jurgens, G.; Knox, J.P.; Wang, M.H. ABA promotes quiescence of the quiescent centre and suppresses stem cell differentiation in the Arabidopsis primary root meristem. Plant J 2010, 64, 764-774, doi:10.1111/j.1365-313X.2010.04367.x.
  2. Ortega-Martinez, O.; Pernas, M.; Carol, R.J.; Dolan, L. Ethylene modulates stem cell division in the Arabidopsis thaliana root. Science 2007, 317, 507-510, doi:10.1126/science.1143409.
  3. Sharp, R.E.; LeNoble, M.E. ABA, ethylene and the control of shoot and root growth under water stress. J Exp Bot 2002, 53, 33-37, doi:DOI 10.1093/jexbot/53.366.33.
  4. Li, Z.F.; Zhang, L.X.; Yu, Y.W.; Quan, R.D.; Zhang, Z.J.; Zhang, H.W.; Huang, R.F. The ethylene response factor AtERF11 that is transcriptionally modulated by the bZIP transcription factor HY5 is a crucial repressor for ethylene biosynthesis in Arabidopsis. Plant J 2011, 68, 88-99, doi:10.1111/j.1365-313X.2011.04670.x.
  5. Zhao, Y.; Xing, L.; Wang, X.G.; Hou, Y.J.; Gao, J.H.; Wang, P.C.; Duan, C.G.; Zhu, X.H.; Zhu, J.K. The ABA Receptor PYL8 Promotes Lateral Root Growth by Enhancing MYB77-Dependent Transcription of Auxin-Responsive Genes. Sci Signal 2014, 7, doi:ARTN ra53

10.1126/scisignal.2005051.

  1. Antoni, R.; Gonzalez-Guzman, M.; Rodriguez, L.; Peirats-Llobet, M.; Pizzio, G.A.; Fernandez, M.A.; De Winne, N.; De Jaeger, G.; Dietrich, D.; Bennett, M.J.; et al. PYRABACTIN RESISTANCE1-LIKE8 Plays an Important Role for the Regulation of Abscisic Acid Signaling in Root. Plant Physiol 2013, 161, 931-941, doi:10.1104/pp.112.208678.
  2. Dietrich, D.; Pang, L.; Kobayashi, A.; Fozard, J.A.; Boudolf, V.; Bhosale, R.; Antoni, R.; Nguyen, T.; Hiratsuka, S.; Fujii, N.; et al. Root hydrotropism is controlled via a cortex-specific growth mechanism. Nat Plants 2017, 3, doi:ARTN 17057

10.1038/nplants.2017.57.

  1. Rezaul, I.M.; Feng, B.H.; Chen, T.T.; Fu, W.M.; Zhang, C.X.; Tao, L.X.; Fu, G.F. Abscisic acid prevents pollen abortion under high-temperature stress by mediating sugar metabolism in rice spikelets. Physiol Plantarum 2019, 165, 644-663, doi:10.1111/ppl.12759.
  2. Zou, J.; Liu, A.L.; Chen, X.B.; Zhou, X.Y.; Gao, G.F.; Wang, W.F.; Zhang, X.W. Expression analysis of nine rice heat shock protein genes under abiotic stresses and ABA treatment. J Plant Physiol 2009, 166, 851-861, doi:10.1016/j.jplph.2008.11.007.

Round 2

Reviewer 2 Report

I still not easy with statement in point 4. If authors feel it ok then may leave it otherwise edit accordingly